# Unsupervised Non-Parametric Signal Separation Using Bayesian Neural Networks

## Abstract

Bayesian neural networks (BNN) take the best from two worlds: the one of flexible and scalable neural networks and the one of probabilistic graphical models, the latter allowing for probabilistic interpretation of inference results. We make one extra step towards unification of these two domains and render BNN as an elementary unit of abstraction in the framework of probabilistic modeling, which allows us to promote well-known distributions to distribution fields. We use transformations to obtain field versions of several popular distributions and demonstrate the utility of our approach on the problem of signal/background separation. Starting from prior knowledge that a certain region of space contains predominantly one of the components, in an unsupervised and non-parametric manner, we recover the representation of both previously unseen components as well as their proportions.

## 1 Introduction

Neural networks as predictive models have been wildly successful across a variety of domains, be it image recognition or language modeling. And while they may be used to make predictions on previously unseen samples, one of fundamental weaknesses of traditional neural networks is the inability to quantify the prediction uncertainty. Evaluation of prediction uncertainty is important in basic research (identification of fundamental laws), reinforcement learning (identification of value functions), anomaly detection, etc.

Uncertainty quantification in neural networks has been addressed both from frequentist (see, for instance, Pearce et al. (2018)) and Bayesian ( Kendall and Gal (2017)) sides. In the Bayesian setting it was naturally proposed to promote the weights of neural layers to normally distributed random variables (MacKay (1992)). Later it was shown that the learnt uncertainty in the weights improves generalization in non-linear regression problems, and it can be applied to drive the exploration-exploitation trade-off in reinforcement learning ( Blundell et al. (2015)). Depeweg et al. (2017) designed a method of separation of uncertainty into epistemic and aleatoric. Epistemic uncertainty expresses uncertainties inherent to the model and can not be reduced with additional observations, whereas aleatoric uncertainty captures the amount of noise due to training on a specific sample. In physics the former and latter are referred to as the systematic and statistical uncertainties, respectively.

In treating both types of uncertainty within the same framework authors essentially bridged the gap towards graphical models. Graphical models, unlike traditional neural networks, are probabilistic in nature and allow for incorporation of prior beliefs with respect to models. They are flexible in representing various processes and allow for introduction of latent degrees of freedom. Initially graphical models used various point distribution as building blocks, while mostly normal distribution has been promoted to a random in the notable example of Gaussian random fields.

In this work we propose using Bayesian Neural Networks (BNN) as building blocks in graphical models and demonstrate the power of synthesis of Probabilistic Graphical Models (PGM) and BNNs on a synthetic example of signal/background separation. As a demonstration of our approach we propose the additive mixture model: a superposition of signal and background spectra whose proportion varies in space. During inference we are able to learn the proportion of signal and background and their spectral shapes that match ground truth values to adequate precision. The paper is organized as follows. In Section 2 we recapitulate the feed-forward (vanilla) BNN and the variational inference approach. In Section 3 we present the transformations of a vanilla BNN that allow to emulate various distribution fields. To illustrate the power of composition of transformed

BNNs in Section 4 we introduce a model of additive BNN mixture. In Section 5 we describe our experiments and in Section 6 we discuss the results and evaluate model performance. In Section 7 we conclude by a discussion of limitations as well as prospective domains of applications of our framework.

## 2 BACKGROUND: VANILLA BAYESIAN NEURAL NETWORK

We consider feed-forward deep neural architectures that are composed of dense layers. A dense layer $k$ is an affine transformation $L^k$ with weight $\mathcal{W}^k$ and bias $\mathcal{B}^k$ that is followed by an element-wise non-linear transformation $\sigma$: $h^k = L^k \circ h^{k-1} = \sigma(h^{k-1}\mathcal{W}^k + \mathcal{B}^k)$, also known as the activation function. In our experiments we set $\sigma$ to be $ReLU$, defined as $ReLU(x) = \max(0, \mathrm{x})$. In what follows we work with a simple linear deep architecture which is defined as a consecutive application (composition) of dense layers: $y = L_K \circ \cdots L_1 \circ x$ .

In order to enable probabilistic interpretation of inference using neural networks, the weights and the biases of each layer are promoted to random variables and are sampled from a Normal distribution with corresponding parameters: $\mathcal{W}^k \sim N(\mu_{\mathcal{W}}, \Sigma_{\mathcal{W}})$, $\mathcal{B}^k \sim N(\mu_{\mathcal{B}}, \Sigma_{\mathcal{B}})$. In Fig. 1, right panel we depict an elementary Bayesian Neural Network, composed of $k$ layers and that takes as input $x$, consisting of $N$ samples, and rendering $y$ as output, using plate notation.

We consider a simple BNN in the spirit of (Blundell et al. (2015)), where authors use stochastic variational inference (SVI) (Hoffman et al. (2013); Wingate and Weber (2013)) for Gaussian posterior distributions from prior distributions of weights, biases and observations. Under these conditions it is natural to use Evidence Lower Bound (ELBO) (Mehta et al. (2019)) as the loss function. ELBO loss consists of two terms: log evidence of the observable variable $x$ with learnable parameters $\theta$, $\log p_\theta(x)$, and the Kullback-Leibler (KL) divergence between the approximation of the posterior distribution $q_\phi(z)$, parametrized by $\phi$, and the true posterior $p_\theta(z|x)$:

$$ELBO = \log p_\theta(x) - \mathrm{KL}\left(q_\phi(z)||p_\theta(z|x)\right). \tag{1}$$

Taking steps in $\phi$ to increase ELBO, increases log evidence and decreases the distance between the prior and the posterior. We further illustrate this in Fig. 1, left. Inference results depend on the choice of the optimizer, the learning rate and the number of iterations.

## 3 TRANSFORMED BNNS

Non-trivial examples of probabilistic models combine distributions of various types. Consider a $K$-component Gaussian Mixture model: each component of the mixture is normally distributed, where the mean parameterized by real-valued parameters and the scale - by a positive parameter, while the overall proportions are sampled from a Dirichlet distribution $X_k \sim Dir(\alpha)$, which, in turn, is parameterized by a positive vector $\alpha_1, \ldots \alpha_K > 0$, and $X_k$ belong to a $K-1$ simplex: $\sum X_k = 1$.

We are therefore motivated to introduce a family of transformed BNNs with various ranges. In this manuscript we consider exponential transformation (transforms unconstrained vector of $K$ dimensions to a positive vector of $K$ dimensions) and a stick breaking transformation (unconstrained vector of $K-1$ dimensions into a simplex vector of $K$ dimensions).

We propose to apply transformations after the last layer of BNN, in such a way that the range of the output is constrained to be strictly positive for the exponential transform and a k-dimensional vector summing to unity (k-simplex) for the stick breaking transformation.

In what follows we denote vector $y$ sampled from a Bayesian neural network as $y \sim BNN(x, (\mathcal{W}, \mathcal{B}))$. We denote $BNN$ outputs transformed by exponential and stick-breaking transforms $BNN_e$ and $BNN_s$, respectively.

Another type of BNN transformation we consider is prompted by probability distributions: in certain applications it is particularly useful to work not just with positive random fields but with normalized positive random fields. Practically such a transformation consists of an approximate normalization of the BNN output given the data.

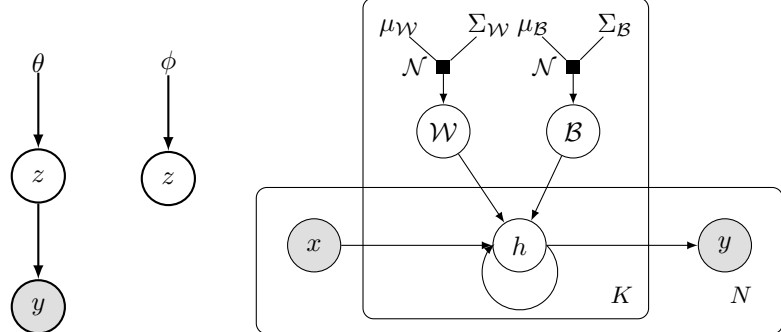

Figure 1: *Left panel*: Graphical model representation of variational inference. Observed random variable is denoted as $x$, latent variables that generate $x$ via $P(x|z)$ are $z$. Prior distribution of $z$ is parametrized by $\theta$: $P(z|\theta)$, the true distribution of $z$ is approximated by a posterior $P(z|\phi)$. The functional form $P(z|\phi)$ may be different from $P(z|\theta)$ (it is important that $P(z|\phi)$ is differentiable with respect to $\phi$). Sampling $z$ from $P(z|\phi)$ and $P(z|\theta)$ allows to evaluate KL divergence term, quantifying distance between them. Meanwhile $z$ sampled from $P(z|\phi)$ is used to evaluate data evidence. In the context of probabilistic programming language *Pyro* $P(z|\theta)$ is part of the "model" and is denoted $p_\theta(z)$, $P(z|\phi)$ is part of the "guide" and is denoted $q_\phi(z)$. *Right panel*: Graphical model representation of an elementary BNN. Shaded circles represent observable variables (input $x$ and output $y$), empty circles - latent variables ($\mathcal{W}$, $\mathcal{B}$ and $h$), standalone letters - hyper-parameters ($\mu_\mathcal{W}$, $\sigma_\mathcal{W}$, $\mu_\mathcal{B}$ and $\Sigma_\mathcal{B}$). Directed arrows represent dependence between the starting and terminal vertices, black squares with $\mathcal{N}$ represent sampling from a normal distribution. Internal plate of dimension $K$ represents $K$ BNN layers and plate of dimension $N$ represents the size of the data sample.

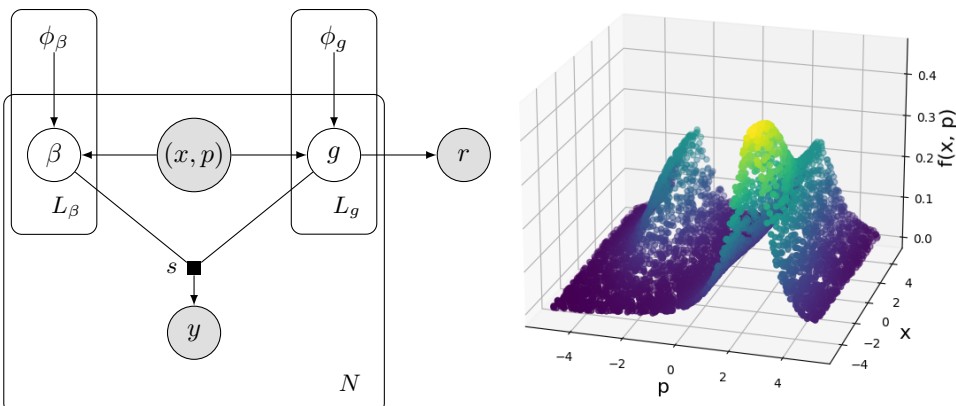

Figure 2: *Left panel*: graphical model representation of additive mixture model. Shaded circles represent observable variables (input $(x, p)$, outputs $y$ and relative entropy $r$), empty circles - latent variables associated with BNNs $\beta$ and $g$, standalone letters - hyper-parameters of BNNs ($\phi_\beta$, $\phi_g$). Directed arrows represent dependence between the starting and terminal vertices, black square $s$ represents the convolution given by Eq. 2. Internal plate of dimension $N$ represents conditional independence of the data sample. Plates $L_\beta$ and $L_g$ represent conditional independence of parameter sampling of $\beta$ and $g$ BNNs. *Right panel*: a sample of 5000 datapoints representing the mixture $f(x, p)$. For negative values of $x$ the total signal is dominated by the background component $g_1(p)$, while at positive values of $x$ the signal and the background are in superposition.

## 4 ADDITIVE MIXTURE

Having introduced architectures with a single dense unit BNN, we turn to a non-trivial test of our framework: a generative additive mixture model. In this model we consider two types of coordinates: we refer to $x$ as the spatial coordinate and $p$ as the spectral coordinate.

We consider two positive spectral functions $g_1(p)$ and $g_2(p)$ that are mixed in a spatially dependent manner by a simplex-valued $\beta(x)$, and suppose that only their sum $f(x, p)$ is observed:

$$f(x, p) = \beta(x)g_1(p) + (1 - \beta(x))g_2(p). \tag{2}$$

This model is motivated by the problem of identification of spectrum of signal $g_2(p)$ in the presence of non-trivial noise $g_1(p)$. We assume that $g_l(p)$ are normalized: $\int dp g_l(p) = 1$. It follows immediately that $f(x) = \int dp\, f(x, p)$ is equal to 1 for each $x$. Spectral functions $g_l(p)$ are positive-valued and we model them by an exponentially transformed $BNN_e$. We model $\beta(x)$ as a Dirichlet random field $\beta(x) \sim Dir(\alpha_l(x))$, where $\alpha_l(x) = N\gamma_l(x)$ and $\gamma \sim BNN_s(x)$ (stick-breaking transformed BNN). In order to deal with degeneracy due to the permutation symmetry: $g_1 \leftrightarrow g_2$ and $\beta \leftrightarrow 1 - \beta$ we set the initial value of $\gamma_l$ to correspond to an asymmetric proportion. e.g $(0.99, 0.01)$, where the first component represents the background, while the other - the signal. To further discourage the inference engine from splitting the observed signal between components uniformly we introduce another observable, the relative entropy between spectral components $g_1(p)$ and $g_2(p)$ :

$$D_{KL}(g_2(p) \parallel g_1(p)) = \int g_2(p) \ln(g_2(p)/g_1(p))$$

and set it to a large number, $e.g.$, 100.

The first observed term forces the combination of $g_l(p)$ and $\beta(x)$ to approximate $y_i$. The term containing relative entropy forces to learn maximally different $g_1(p)$ and $g_2(p)$, since $D_{KL}(g_2(p) \parallel g_1(p))$ is a proxy to distance in function space.

The objective of the inference is to identify optimal model parameters $\phi_\beta$ and $\phi_g$ and thus to obtain the full spatial and spectral description of the mixture, $i.e.$, the shapes of $g_l(p)$ and the mixing proportion $\beta(x)$. In the current formulation the additive mixture model does not include Bayesian treatment aleatoric uncertainty, observations are sampled using a small fixed variance (0.002).

## 5 EXPERIMENTS

BNN abstractions and experiments were implemented[1] in Pyro (version 1.5.2) (Bingham et al. (2019)), a probabilistic programming language (PPL) written in Python and based on *pytorch*, which enables Bayesian probabilistic modeling thanks to Monte Carlo and variational inference engines. In our experiments we use unit BNNs containing 3 hidden layers of dimensions $32 \times 128 \times 32$.

**Inference**. We use Pyro's stochastic variational inference (SVI) abstraction which computes ELBO loss and take steps in the space of "guide" parameters $\phi$ along the gradients of loss function. As a whole the procedure is a Bayesian update: it identifies variational parameters $\phi$ of the true posterior approximation $q_\phi(z)$. In what follows we use Clipped Adaptive Moment Estimation (ADAM) (Kingma and Ba (2014)) as the optimization method and, unless mentioned otherwise, set the learning rate to $10^{-2}$, the clipping norm to 10, and default values for the coefficients used for computing running averages of gradient and its square: (0.9, 0.999).

The value of the ELBO loss is computed for each epoch. The relative values of the loss with respect to the previous iterations is computed too. For a chosen window, the former are compared with two conditions.

We improve the inference procedure in an empiric manner by restarting it with a decreasing value of learning rate (factor of 0.5 in our case), while keeping current "guide" parameters. The restart happens if $(i)$ the last $S_p$ steps resulted in an increased loss or $(ii)$ the last $S_r$ steps resulted in a relative loss below a certain threshold. Both $S_p$ and $S_r$ are chosen to be 3 in the current project. Condition $(i)$ means that the inference engine is diverging from optimal region of the parameter space (likely due to the stochastic nature of sampling), which manifests in the divergence of loss function. This procedure is repeated until a fixed number of steps are performed.

**Training of unit BNNs**. Unit vanilla and transformed BNNs introduced in Section 3 may be used to approximate functions or as latent constituents of more complex models. When using them as latent parts of more complex models it is often advantageous to impose certain priors on their functional

---

[1]Implementation available at `https://github.com/repo_url`.

form, $e.g.$, we might want to represent the prior knowledge of mixture composition, which is not directly observed.

In other words latent components of a complex model can be pre-trained so that their priors condition the output to have the desired shape. If the prior and the "guide" have the same functional form, this procedure can be viewed as a Bayesian update. Using learned during pre-training "guide" parameters to initialize the "guide" when a given unit BNN is used as a constituent may improve convergence.

In short we go through the following steps: (i) generate a data sample emulating desired shape, (ii) fit a unit BNN to the generated data sample using non-informative priors, thus obtaining "guide" parameters, and (iii) use inferred guide parameters as prior parameters and (optionally) as the initial "guide" parameters as part of a more complex model. The second step involves sampling BNN output to match observations. Since we are not interested in aleatoric component at this stage, we represent it as a constant and use normal, log-normal and Dirichlet for sampling.

We illustrate the process of inference of a unit BNN on the example of approximating $g^\star(p) \sim e^{\frac{1}{1+p^2}} - 1$, shifted by $b = -2.5$ (and normalized to 1). To fit this positive function we use an exponentially transformed $BNN_e$, which is sampled using log-normal distribution after the last layer by convention. The inferred posterior distribution of the inference on a unit $BNN$ is shown in the left panel, the loss - in the second-to-left, the location parameters $\mu$ of weights $\mathcal{W}$ of the last hidden layer of given BNN a function of epoch - in the second-to-right and the evolution of scale parameters $\Sigma$ - in the right panel of Fig. 3. The inferred posterior distribution matches well the true values within the 68% containment bands of the model.

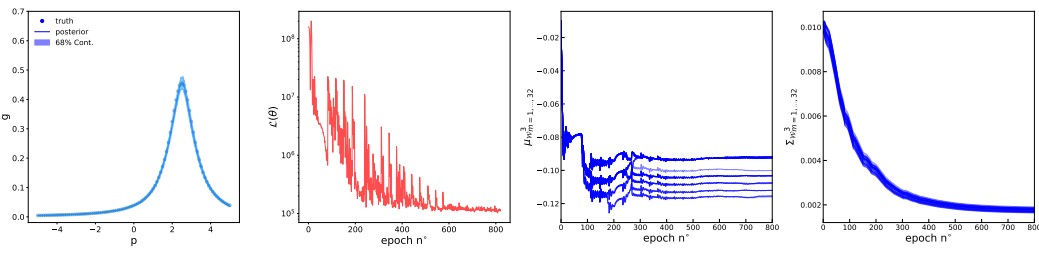

Figure 3: *Left panel:* true versus predicted values from a unit $BNN_e$. Mean value is shown as solid line and the 68% containment bands as the shaded area. Solid circle represent sampled true values. *Second-to-left panel:* ELBO loss function as a function of training epoch for a unit $BNN$. *Second-to-right panel:* posterior location hyperparameters of the weight distribution $\mathcal{W}$ for 32 neurons of the last hidden layer of a unit $BNN$ as a function of training epoch $n$. The location parameters of $\mathcal{W}$ are initialized to zeros in the beginning of SB inference. *Right panel:* posterior scale hyperparameters of the weight distribution $\mathcal{W}$ for 32 neurons of the last hidden layer of a unit $BNN$ as a function of training epoch $n$. Scales are initialized to 0.1 in the beginning of SB inference.

**Synthetic Additive Mixture**. By considering the additive mixture model we answer the following fundamental question: if a superposition of two spectral ($i.e.$ positive) functions is observed as a function of spatial coordinates, can we learn their shapes and proportions, given a minimum amount of information? To test our method for the additive mixture model we create a synthetic dataset containing triples $\mathcal{D} = \{(x_i, p_i, f_i)\}_{i=1}^N$, $x_i \in \mathcal{R}$, $p_i \in \mathcal{R}$, where $x_i$ and $p_i$ are sampled uniformly on $[-5, 5]$ and refer to this particular implementation as the Signal/Background (SB) model. The values of $f(x_i, p_i)$ are derived using the generative model of Sec. 4 with deterministic $g_l(p)$ and deterministic $\beta(x)$. In what follows we refer to $g_1(p)$ and $g_2(p)$ as the background and the signal, respectively. We consider (i) case A where non-informative priors are used, and (ii) case B where only $g_1(p)$ is conditioned. For both of them we use a previously introduced template $g^*(p)$ (normalized to 1), which we shifted by $b = 2.5$ in two directions: $g_1(p) = g^*(p - b)$ and $g_2(p) = g^*(p + b)$.

First we demonstrate that the inference works for a specific choice of $\beta$: we choose $\beta(x)$ to be a logistic function with values at $-\infty$ equal to 0.99 and $+\infty$ equal to 0.5. Afterwards we test the limit of our approach by considering synthetic datasets with decreasingly small proportion of "signal".

$\beta(x)$ and $g_l(p)$ are represented by a stick-breaking and an exponentially transformed BNNs (see Sec. 4), hereafter referred as to $BNN_\beta$ and $BNN_g$, respectively. Both of them have output dimensions

equal to 2. The prior distributions for $(\mathcal{W}, \mathcal{B})$ of all hidden layers of $BNN_\beta$ and $BNN_g$ are taken to be $\sim N(0, 0.1)$.

It is instructive to represent $BNN_\beta(x, \mathcal{W}, \mathcal{B})$ graphically as a function of $x$, where $\mathcal{W}, \mathcal{B}$ are sampled from prior distributions: in Fig. 5 the mean values are plotted as dashed lines together with the 68% containment bands as shaded areas for case A and case B in the top and bottom panel, respectively.

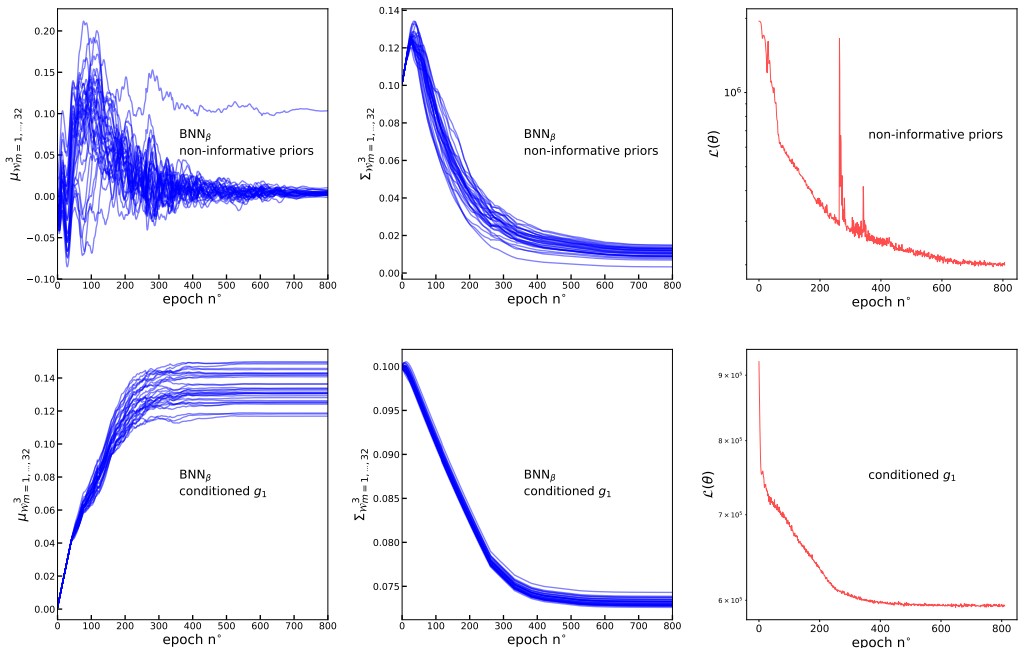

Figure 4: *Left panel:* posterior location hyperparameters of the weight distribution $\mathcal{W}$ for 32 neurons of the last hidden layer of $BNN_\beta$ as a function of training epoch $n$. The location parameters of $\mathcal{W}$ are initialized to zeros in the beginning of SB inference. *Middle panel:* Posterior scale hyperparameters of the weight distribution $\mathcal{W}$ for 32 neurons of the last hidden layer of $BNN_\beta$ as a function of training epoch. Scales are initialized to 0.1 in the beginning of SB inference. *Right panel:* ELBO loss function as a function of training epoch. For left, center and right panels top figures are given for the case non-informative prior (top panels) and the case of the prior, pre-trained on the background spectral shape $g_1(p)$ (bottom panels).

Location parameters of the $\mathcal{W}$ for the last hidden layer of $BNN_\beta$ as a function of epoch are shown in the left panel of Fig. 4. The scale parameters are shown in the middle panel of the same figure. The right panel shows the ELBO loss as a function of epoch. The top panels show the results for case A and the bottom panels for case A. For the case of non-informative prior the variation of the means is negligible after 700 epochs. In the case where the $g_1$ is conditioned, locations hyperparameters converge to their terminal values after 500 epochs.

## 6 RESULTS

In what follows, we set $\beta(x)$ to vary continuously from $\beta^T|_{x=-5} = 0.99$ in the background-dominated spatial region to $\beta^T|_{x=5} = 0.7$ in the mixture region.

The inferred posterior distribution of $1\text{-}\beta(x)$ and $g_l(p)$ are shown in Fig. 5 for proportion in the mixture region of $\beta^T|_{x=5} = 0.7$ for cases A and B in the upper and lower, left and second-to-left panels, respectively. The inferred posterior distribution for $1\text{-}\beta(x)$ matches well the true values within the 68% containment bands of the model for cases A and B.

For case B, the relative error between the true signal proportion at $x = 5$, *i.e.* $1\text{-}\beta^T|_{x=5}$, and the mean proportion recovered by the model is lower than 8% and the prediction of $g_2$ is in agreement

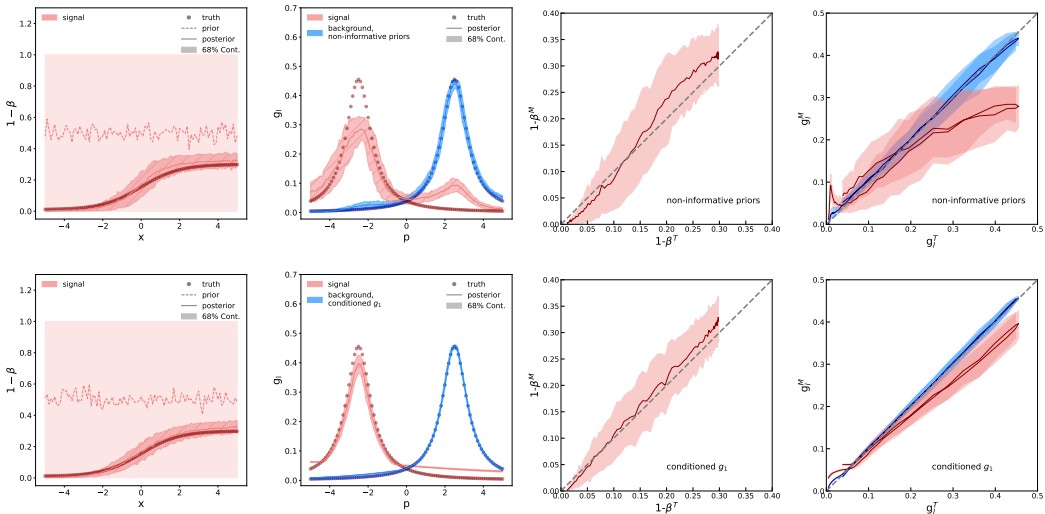

Figure 5: Predictions compared to true values of signal proportion $1 - \beta$ (left) and spectral shapes $g_l$ for background and signal obtained from the SB inference, in blue and red respectively. True values of $\beta^T|_{x=5} = 0.7$ are assumed. *Left panels:* comparison between predictions and true values for the mixing signal proportion $1 - \beta(x)$. The mean is shown as solid line and the 68% containment bands as the shaded area. Solid circle represent sampled true values. Predictions from prior distributions are represented by their means (dashed line) and 68% containment bands (light shaded area). *Second-to-left panels:* comparison between predictions and true values for the spectral shapes $g_l(p)$. *Second-to-right panels:* model signal proportion $1 - \beta^M$ vs the true proportion $1 - \beta^T$ together with 68% model uncertainty extracted from the left panels. *Right panels:* model spectral shapes $g_l^M$ vs true spectral shapes $g_l^T$ together with their 68% uncertainty extracted from the second-to-left panels. Top and bottom panels show results for case A with non-informative priors and case B with conditioned $g_1(p)$, respectively.

with true values within 95% confidence level. For case A, the relative error between the true signal proportion and the mean proportion recovered by the model is lower than 9% but the ground truth values of $g_2$ are not contained in the 95% confidence band.

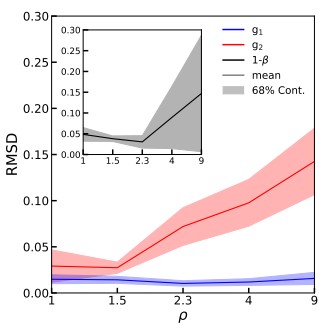

Figure 6: RMSD for the comparison of the predictions after the SB inference and the true values of $1$-$\beta$ and $g_l$ vs the noise-to-signal ratio $\rho$.

The second-to-right and right panels of Fig. 5 show the so-called y-y plot of predicted versus the ground truth values for the signal proportion $1$-$\beta$ and the spectral shapes $g_l$ for the cases A and B in the top and bottom panels, respectively. The uncertainties are given as 68% containment bands (shaded areas). The true values of the proportion are recovered by the model at 68% containment level, in both cases. In case B, biases from 1% to 5% are obtained for the model signal proportion. For the spectral shape $g_2$, a bias up to 16% is obtained for $g_2^T$ values larger than 0.1. In case A, biases from 1% to 10% are obtained for the signal proportion. For the spectral shape $g_2$, a bias up to 39% is obtained for $g_2^T$ values larger than 0.1.

We also run a series of experiments by changing $\rho = \frac{\beta^T(x=5)}{(1-\beta^T(x=5))}$ in the ground truth data, or, in other words, the ground truth background fraction value at the right boundary, which may be interpreted as the noise-to-signal ratio. Each experiment is repeated several times with random seeds to help us mapping the distribution. In Fig. 6, we plot the root mean squared deviation (RMSD) between the prediction of the corresponding constituent BNN after the SB inference and the true value of $1$-$\beta$ and $g_l$, respectively, as a function of $\rho$. We note that the variation

of RMSD for $g_1$ over the whole range of $\rho$ is small. We also note that the error in identification of $g_2$ and 1-$\beta$ increases with increase of $\rho$, potentially due to the fact that the inference procedure is no longer able to distinguish a small signal from the fixed aleatoric noise.

## 7 DISCUSSION

In this work we introduced a framework that merges the powers of Bayesian neural networks and graphical models and demonstrated its validity on a proof of concept model of signal/background separation. This framework allows to compose BNNs, representing random fields, in the same way point distributions are composed in graphical models.

Our proof of concept model is motivated by the astrophysical problem of learning of an unknown signal in the presence of a potentially unknown background, which manifests at various wavelengths. We lift this traditionally treated by parametric statistical methods and more recently using Bayesian statistics (see, for instance, Abdallah et al. (2016; 2018; 2020); Abdalla et al. (2021)) problem, thanks to the power of neural networks, to a new level of non-parametric inference. Our approach is non-parametric with respect to proportions and spectral shapes and also Bayesian; it allows for Bayesian updates: to incorporate available extra information and to aggregate spectral descriptions by class (study sources that we believe belong to the same class).

We outline several possible improvements and generalizations that will be addressed in future work.

- At the heart of our framework of representing random fields by BNNs is a non-linear transformation of a neural network layer, that renders the output positive. According to Jensen inequality increasing the scales parametrizing a BNN will lead to a greater expectation value of the output (as well as hidden layers). This might lead to a potential problem during inference using ELBO loss, where matching expected values (of potentially latent random variables) might be achieved by increasing the scale parameter, instead of increasing the location parameter, at the same time rendering log probabilities for the sampled distribution irrelevant (as in any data fits equally well a very wide Gaussian).

- From the point of view of physical modeling of signal/background separation, our model is built upon several simplifications: it does not include an explicit model of the measurement, nor does it contain a microscopic emission model. It also posits space-energy factorization, that energy-spectra are spatially independent, which is not the case if the target (gas, radiation) density fields are strongly spatially-dependent. The gamma-ray signal expected from cosmic ray interaction in the interstellar medium would therefore exhibit a spatially-dependent spectral behavior. All the above points maybe address within the current framework.

- In SB model we considered the separation of total signal into two components in one spatial dimension. In astrophysics and cosmology raw observations are made in two spatial coordinates and time. Our framework allows very easily to extend the number of spatial dimensions. Increasing the number of components on the hand should be done with caution as it decreases the stability of inference due to appearance of extra permutation symmetries.

- The additive mixture example was focused on identification of epistemic uncertainties. Aleatoric uncertainty may be represented as a parameter or another unit BNN.

- SB model was demonstrated to apply in a limited signal-to-noise ratio region. While this might be due to our choice of fixed aleatoric scale, it is possible that inference using ELBO loss becomes less stable, when smaller scales are chosen.

SB model may be directly applied in gamma-ray, cosmic-microwave-background and gravitational astrophysics. The framework, however, is far more general and could be used, in the context of Non Intrusive Load Monitoring (NILM), where the total household consumption signal is disaggregated into multiple house appliances and over-fitting problem is of particular importance (Jones et al. (2020)). Other applications include Bayesian modeling of value function in reinforcement learning (Eriksson et al. (2020)) and audio source separation (Schulze-Forster et al. (2022)).

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
