# OpenReview forum: "Unsupervised Non-Parametric Signal Separation Using Bayesian Neural Networks"
_ICLR.cc/2023/Conference — Submitted to ICLR 2023_

### Official Review · Reviewer_6vPR · 2022-10-17

**Confidence:** 4
**Correctness:** 3
**Technical Novelty And Significance:** 3
**Empirical Novelty And Significance:** 2
**Recommendation:** 3

**Clarity, Quality, Novelty And Reproducibility:**


Questions to clarify

1. In Figure 2, the relative entropy r is treated as an observed variable. I found this confusing.
2. What is the observation model for y in the additive mixture model? At the end of Section 4, it says “the observations are sampled using a small fixed variance (0.002)”, but it would seem a little bit inconsistent if the intensities are assumed to be non-negative.
3. How to balance the ELBO term and relative entropy term in the objectives?
4. Is the citation to “gradient clipping” precise?
5. What exactly is the stick-breaking transformation used? Citations missing? Is there any implicit bias in constructing a simplex vector in this way?
6. The "training of unit BNNs" part reads very confusing. "(i) generate a data sample emulating desired shapes (ii) fit a unit BNN...obtaining guide parameters (ii) using the "inferred guide parameters" as prior for more complex model". What are the motivation and benefits of these procedures?


**Strength And Weaknesses:**

It is an interesting idea to study combining Bayesian neural networks with probabilistic graphic models, and get the best of both worlds.  However, the current paper seems not in a form that can be accepted yet. The authors also quite frankly listed a few limitations of the current approach in Section 7.

There are many aspects that require further development, listed below,

From the probabilistic modeling point of view, it is unclear what is the benefit of bringing in the Bayesian neural network piece, compared to other conventional choices such as Dirichlet prior or Gamma vectors, or log-normal vectors.  The potential advantage might include being easier to incorporate covariates, with more flexible dependencies, and/or nonlinearity from neural networks, but this requires more supporting evidence.

The model is tied to a variational model in Pyro, inference is conducted by maximizing the ELBO with an additional penalty term of relative entropy to encourage the separation of components. A key question unaddressed: how does the proposed framework compare to the standard VAE framework, with a probabilistic graphical model as the generative model, and NN or transformed NN as the recognition model?

From the application point of view, it is unclear how the simple additive mixture model is compared to other choices of non-parametric models such as Gaussian processes.  The non-parametric aspect for unsupervised learning is especially interesting. Is it possible to explicitly show some differentiation achieved with the proposed approach?

The argument of building a random field rather than point distributions also needs more development, e.g., what can be achieved beyond Gaussian random fields?  Although a few potential applications are listed in Sec. 7., the paper would certainly benefit from evaluations on some real-world datasets.


**Summary Of The Paper:**

This paper proposes to use parametric transforms such as exponential function or stick-breaking transformation on the outputs of Bayesian neural network (BNN) and turn them into positive random vectors or simplex vectors. This new transformed BNN module can be plugged into probabilistic graphic models as a replacement prior. Sampling from transformed BNN is also possible, with the priors on the weights and biases. An additive mixture model is introduced for signal background separation with synthetic experiments.

**Summary Of The Review:**

Overall, I think this paper needs further development to be accepted, including both a more formal justification of the proposed framework, comparisons to other methods such as VAE or GPs models,  and experimental validations on real-world datasets.

---

> ### Author Response · Authors · 2022-11-19
> **response**
>
> We thank reviewer for carefully reading our submission and pointing out the deficiencies on which we hope to improve in future.
>
> We considered the introduction of Bayesian neural networks due to their ability to describe continuous functions thanks to universal approximation theorem, unlike Dirichlet priors, describe only a subclass of continuous functions. We agree with the reviewer that the utility of using non-parametric estimators (for spectral shapes, for example) should be shed light upon.
>
> If we understand correctly the reviewer suggests using a VAE that encodes the total signal and then decodes it using the graphical model. We agree that the our approach should be benchmarked against a standard framework, however, we have not found successful applications of VAE to additive mixture in an unsupervised setting.
>
> We agree with reviewer that the non-parametric aspect of the model should be highlighted and contrasted with a parametric benchmark. We will be working to improve this in a future submission.
>
> 1. In the additive mixture model we introduce relative entropy between spectral functions $g_i(p)$ as an observed random variable. Since $g_i(p)$ are represented by Exp-transformed BNNs, we sample from $g_1(p)$ and $g_2(p)$ in order to compute KL divergence. In that sense relative entropy is a function of random variables. Although in real-life relative entropy is not observed, we use this term to make $g_1$ different from $g_2$, akin to regularization. Due to its function the precise value of “observed” relative entropy is not important, as long as it’s somewhat large. It corresponds an empiric prior assumption that the shape of signal is different from the shape of the background. We tried to clarify this in the main text.
> 2. We agree with the reviewer that sampling a positive random variable with Normal dist with finite variance might lead to negative values. In this passage we are actually talking about the log of probability of y_obs given y using a normal distribution, so practically it does not lead issues. However, we agree that conceptually we should be using a different type of distribution.
> 3. Indeed there is a possible interplay between the ELBO term and the relative entropy, however, since we explained that the relative entropy term observed value is somewhat arbitrary if large enough, the balance is mediated by its value.
> 4. The citation refers ADAM, rather than to gradient clipping.
> 5. The stick breaking transform is the transform of unconstrained space to the simplex of one additional dimension via a stick-breaking process as discussed:
> https://github.com/pytorch/pytorch/blob/master/torch/distributions/transforms.py#L899
> https://mc-stan.org/docs/2_18/reference-manual/simplex-transform-section.html
> 6. This procedure allows to condition the constituent BNNS of the PGM to sample outputs respecting our prior expectations. So for example if we know that a certain region of space is unlikely to contain signal, we can pre-train and condition the BNNs denoting signal/noise proportions to reproduce the fraction of noise as something very small.

---

### Official Review · Reviewer_YQeZ · 2022-10-24

**Confidence:** 4
**Correctness:** 3
**Technical Novelty And Significance:** 1
**Empirical Novelty And Significance:** 1
**Recommendation:** 1

**Clarity, Quality, Novelty And Reproducibility:**

The main idea of the paper seems to be replacing standard distributions within a PGM with Bayesian neural networks whose outputs satisfy the properties of a distribution, to obtain some sort of non-parametric alternatives for conditional distributions. This standard approach is described as a novel idea and it is decorated with some additional terminology ('transformed BNN' that seems to mean standard BNN with specific activation function for the output layer), but I still fail to see where the actual contribution of the paper lies. Probabilistic programming libraries, like Pyro that the authors also use, are specifically designed to make building this kind of models easy and using neural networks as function approximations within those models is a routine practice. The idea itself is hence by no means bad, but it also has no novelty and the basic idea of using neural networks within a PGM and variational inference dates at least back to 2005 (Harva et al. "Bayes blocks: an implementation of the variational Bayesian building blocks framework", UAI 2005). The case study in Section 4 is something most Pyro users should be able to write if given the task and the solution they would build is roughly the same that is proposed here (including even the choice of the inference method), to the extent that it could perhaps even be used as an exercise on a MSc level course.

The story is also a bit problematic. I think the authors are building towards a general-purpose approach, but in practice only present two very simple examples (stick-breaking and exponentiation) and then proceed to study a very simplified case example. The paper would be more interesting if more challenging examples (perhaps normalising flows?) were characterised as the building blocks and the case-study would be for a problem that is clearly challenging and could not be easily addressed with the standard tools.

**Strength And Weaknesses:**

Strengths:
- The basic approach is sound and well in line with the modern literature where flexible function approximations are being used to expand standard model families

Weaknesses:
- The specific contribution is unclear and the paper looks more like a case study of a well-known principle, where the case being studied is also quite simple
- The writing is a bit naive and known concepts (treating neural networks as probabilistic elements) are introduced as if they were new

**Summary Of The Paper:**

The authors incorporate Bayesian neural networks as non-parametric alternatives for standard distributions within a probabilistic graphical model, and demonstrate the approach in a toy example of separating two independent signals from a mixtures.

**Summary Of The Review:**

Extended case study of how probabilistic graphical models can use neural networks within the model, assuming they are implemented using the standard libraries.

---

> ### Author Response · Authors · 2022-11-19
> **response**
>
> We agree with the reviewer that activation functions and transformations in general play the same role, with a small difference that it is usually implied that activation functions act on an NN layer element-wise, while transformations may mix nodes from a layer and change the dimensionality of output, such as the stick-breaking transform, which maps unconstrained space to the simplex of one additional dimension, hence we prefer to use the term “transformed”.
>
> We thank the reviewer for the relevant reference. We completely agree that the idea of “replacing standard distributions within a PGM with Bayesian neural networks whose outputs satisfy the properties of a distribution” has been discussed in the literature and admit that we failed to find the relevant references at the time of writing, which affected the style of the paper.
>
> The reason we embarked upon this project is precisely this : we could not find ready-made PPL tools to express a problem like signal/background separation available. And while to setup inference of a single BNN is a fairly simple exercise in coding, combining several of them is less trivial. While we take as a compliment the suggestion that it could be used “as an exercise on a MSc level course”, since we believe that MSc courses should be on the cutting-edge, the code we developed could potentially become part of Pyro and save days, if not weeks of work to applied researchers combining BNNs as building blocks.
>
> We agree with the reviewer that more challenging examples should be considered and are planning to present such application in the future versions of the draft. We would like to kindly ask the reviewer to recommend a standard Bayesian method to solve the additive mixture problem we consider in the submission in an unsupervised non-parametric setting.
>
> We thank reviewer for carefully reading our submission and pointing out the deficiencies on which we hope to improve in future.

---

> > ### Comment · Reviewer_YQeZ · 2022-11-21
> > **Acknowledging response**
> >
> > Thank you for the response.
> >
> > > The reason we embarked upon this project is precisely this : we could not find ready-made PPL tools to express a problem like signal/background separation available. And while to setup inference of a single BNN is a fairly simple exercise in coding, combining several of them is less trivial.
> >
> > Just to clarify my point: I full agree that building the specific model you designed is not trivial and it certainly takes effort, especially in implementation and experimentation to make it work well and efficiently. In other words, I agree that your work has value. However, it is simply quite clearly below the threshold of what should be published in the top venue -- these venues are for introducing new scientific concepts, ideas or advances algorithms (in the case of CS venues), and your work does not attempt to do that. It would be a great basis for a tutorial of some kind, and most likely is publishable in a less selective venue, especially if coupled with an interesting application.

---

### Official Review · Reviewer_SrCA · 2022-10-25

**Confidence:** 5
**Correctness:** 1
**Technical Novelty And Significance:** 1
**Empirical Novelty And Significance:** Not applicable
**Recommendation:** 1

**Clarity, Quality, Novelty And Reproducibility:**

- Clarity: The paper is simple and clear.
- Quality: The method is too simple.
- Novelty: All methods mentioned in the paper are known in the literature.
- Reproducibility: The URL to GitHub doesn't work.

**Strength And Weaknesses:**

## Strength
- The paper is easy to follow

## Weakness
- The techniques mentioned in the paper are not new and the contributions are over-claimed.
    - Having a transformation in the output layer of neural network or for any probability distribution is a common technique used in the community. Both transformations used in the paper are standard.
    - Additive mixture is another common trick, in both deep learning community and graphical modeling community.
    - Inference method used (SVI) is also standard.
- The experiment is too simple and if signal separation is the focus, some real-world datasets that used for benchmarking purpose should be considered.

**Summary Of The Paper:**

The paper proposes to add non-linear transformation to Bayesian neural networks and uses them in an additive mixture model. They should that this can solve signal separation problems on 2 synthetic tasks.

**Summary Of The Review:**

The proposed method is not novel at all and the contribution is over-claimed. The empirical study is also solely based on simple simulated data.

---

> ### Author Response · Authors · 2022-11-19
> **response**
>
> We thank the reviewer for carefully reading our submission. We agree with the fact all the constituents of our framework are standard. We build upon an existing Probabilistic Programming Language Pyro, which standardized the construction of PGMs in the variational setting. We are not aware, however, of such an application of bayesian neural networks and graphical models to solve an additive mixture in literature and would like the reviewer to kindly share relevant references. We agree that the project is incomplete without benchmarking our method against real-world datasets and would like to kindly ask the reviewer if she/he had any specific benchmarking datasets in mind. Perhaps, one the virtues of our work is the software extension of Pyro PPL, that allows for using BNNs as units of abstraction.
>
> The code was provided at the time of the submission as a zip archive in compliance with  “Anonymous Url” ICLR policy. The URL was intentionally left invalid for the same reason.

---

> > ### Comment · Reviewer_SrCA · 2022-12-02
> > **Acknowledging author response**
> >
> > Thanks for your response.
> >
> > > We are not aware, however, of such an application of bayesian neural networks and graphical models to solve an additive mixture in literature and would like the reviewer to kindly share relevant references.
> > > We agree that the project is incomplete without benchmarking our method against real-world datasets and would like to kindly ask the reviewer if she/he had any specific benchmarking datasets in mind.
> >
> > There are many projects that apply some specific techniques to solve a specific problem. I would agree that your work has its own value. However, having a unique combination of such doesn't mean it passes the bar of NeurIPS, especially when the choices themselves don't bring any new/deeper insight to the ML community.
> >
> > If the solution shows a big progress in the specific problem domain, the author may still want to have it published in a domain-specific venue. However, the empirical results in the paper depend only on synthetic data, and I'm not an expert here to judge what empirical results are considered as strong enough in the signal separation domain.
> >
> > > Perhaps, one the virtues of our work is the software extension of Pyro PPL, that allows for using BNNs as units of abstraction.
> >
> > If this is the case, the presentation of the paper should be completely changed, and the focus should be switched from how to use such abstraction for solving signal separation to how such abstraction can be useful in general, what's the model family it automates (and why it's important), etc.
> >
> > > The code was provided at the time of the submission as a zip archive in compliance with “Anonymous Url” ICLR policy. The URL was intentionally left invalid for the same reason.
> >
> > I believe what mean by "can be downloaded via an anonymous URL" in the policy is that you provide an URL that is anonymous (e.g. via https://anonymous.4open.science/) but still works.
> >
> > Given the reason above I will keep my current score.

---

### Official Review · Reviewer_CEDA · 2022-10-25

**Confidence:** 4
**Correctness:** 3
**Technical Novelty And Significance:** 2
**Empirical Novelty And Significance:** 2
**Recommendation:** 3

**Clarity, Quality, Novelty And Reproducibility:**

The paper is well motivated and easy to follow.
Yet it hides some detail that are important for understanding the contribution. In particular, it overpromises by stating to unify the domains of BNNs and PGMs, while elegantly omitting some of the underlying assumptions.
I understand that this is a hard problem for which it is generally not straightforward to minimize the KL divergence. At the very least, the paper could explain this and make an argument for their assumptions.

The paper also has a very slim list of references with respect to work that has been done on the intersection of those two domains. For example, it does never mention the work of [1] that considers a similar setting.

[1] Johnson, Matthew J., et al. "Composing graphical models with neural networks for structured representations and fast inference." Advances in neural information processing systems 29 (2016).

**Strength And Weaknesses:**

The considered problem of combining data-driven models with probabilistic models is clearly a timely one. Using Bayesian neural networks in that context promises to provide a seamless integration because of the joint probabilistic interpretation. While I would really like to see some fundamental progress being made in that direction, I am afraid that this paper does not do so. The main problem I see is that
the authors simply put those two parts together without accounting for the subtleties that usually arise when considering practical scenarios. This does not become too obvious as the paper makes some strong implicit assumptions as e.g., the consideration of simple mixture distributions with few components and experiments with well separable signal and background noise.

There are also some technical inaccuracies that are sloppy at best and might suggest that the authors did not put sufficient work into understanding the probabilistic graphical models framework at worst. For example, in Figure 1 it is written that the observed random variable is denoted as x with a latent variable z, i.e., we have the underlying data generating process $P_\theta (x|z)$. Yet the left figure clearly corresponds to a data generating process of  $P_\theta (y|z)$.

Similarly, in Figure 2 the relative entropy is described (and illustrated) as a random variable, although it is clearly a function thereof.

**Summary Of The Paper:**

The incorporates Bayesian neural networks into probabilistic graphical models and apply their model to signal/noise decomposition.

**Summary Of The Review:**

The paper tackles an interesting problem of combining data-driven models with probabilistic ones. The contribution of the paper is limited in that it makes strong assumptions on the probabilistic model and does not discuss how both aspects can be merged together effectively in a more general setting. Also, the authors seem to miss some important prior work on that topic.

---

> ### Author Response · Authors · 2022-11-19
> **response**
>
> We thank reviewer for carefully reading our submission and pointing out the deficiencies on which we hope to improve in future.
>
> We corrected left panel of Fig.1 to match the caption.
>
> In the additive mixture model we introduce relative entropy between spectral functions $g_i(p)$ as an observed random variable. Since $g_i(p)$ are represented by Exp-transformed BNNs, we sample from $g_1(p)$ and $g_2(p)$ in order to compute KL divergence. In that sense relative entropy is a function of random variables. Although in real-life relative entropy is not observed, we use this term to make $g_1$ different from $g_2$, akin to regularization. Due to its function the precise value of “observed” relative entropy is not important, as long as it’s somewhat large. It corresponds an empiric prior assumption that the shape of signal is different from the shape of the background.
>
>
> Indeed inference of PGMs using variational methods is hard due to the stochastic nature of loss surface. We reworked the text to highlight the fact that we 1) build upon the probabilistic programming language Pyro to compute ELBO-loss and make steps in the parameter space and 2) using small value of variances for posteriors to facilitate the optimization process.
>
>
> We thank the reviewer for a valuable reference and escaped our attention and expand the relevant section by including several others.

---

> > ### Comment · Reviewer_CEDA · 2022-11-22
> > **Acknowledge response**
> >
> > I have read the response. I agree with reviewer YQeZ that the implementation is not necessarily trivial; still, the paper contains no real methodical contribution that would be a definite requirement for a paper at this venue. Moreover, given that there was a large body of literature being neglected by the authors, a substantial revision of the paper will be required in any case.

---

### Comment · Area_Chair_4A2C · 2022-11-19
**Please respond to author feedback**

Dear reviewers,

The authors have provided their feedback. Please respond, and at least acknowledge you've read them.

Best, AC

---

### Decision · Program_Chairs · 2023-01-20

**Decision:**

Reject

**Justification For Why Not Higher Score:**

Reviewers all favour strong rejects due to significant novelty and clarity issues.

**Justification For Why Not Lower Score:**

N/A

**Metareview: Summary, Strengths And Weaknesses:**

This paper describes a type of Bayesian neural network and applied this model to signal/noise decomposition.

While reviewers acknowledged the practical application is interesting, however they also agree that the proposed approach has very little technical novelty, and the experimental results are not significant enough to be published at ICLR.

I agree with the reviewers and I recommend the authors to consider submitting to venues considering ML applications mainly.


**Summary Of Ac-Reviewer Meeting:**

N/A